# Diet of Broilers with Essential Oil from *Citrus sinensis* and *Xylopia aromatica* Fruits

**DOI:** 10.3390/ani13213326

**Published:** 2023-10-26

**Authors:** Marcela Christofoli, Weslane Justina da Silva, Nathan Ferreira da Silva, Nadielli Pereira Bonifácio, Christiane Silva Souza, Fabiano Guimarães Silva, Paulo Sérgio Pereira, Cibele Silva Minafra

**Affiliations:** 1Goiano Federal Institute of Education, Science and Technology (Federal Institute Goiano–IF Goiano), Rio Verde 75901-970, GO, Brazil; christofolimarcela@gmail.com (M.C.); nathan_zootec2017@outlook.com (N.F.d.S.); nadielli@yahoo.com.br (N.P.B.); fabiano.silva@ifgoiano.edu.br (F.G.S.); paulo.pereira@ifgoiano.edu.br (P.S.P.); 2Goias Federal University, Goiânia 74690-631, GO, Brazil; weslanejds@gmail.com; 3Institute of Animal Science, Federal Rural University of Rio de Janeiro, Rio de Janeiro 23897-000, RJ, Brazil; christianessouza@gmail.com

**Keywords:** limonene, orange essential oil, poultry, phytogenic additives

## Abstract

**Simple Summary:**

The prohibition of the use of antibiotics in the diet of animals occurs because these compounds selectively stimulate the growth of intestinal microbiota among animals raised for human food. This creates health problems among these animals. In addition, their use increases the risk of antibiotic drug resistance among humans who consume the animals; these drugs are important for human health, and resistance builds over time due to the presence of residues of these antibiotics. Faced with this context, research is being published every day on the search for alternative phytogenic additives for animal nutrition; for instance, in the case of poultry, research has been conducted on the use of high-quality ingredients for chicken feed, so that higher-quality meat is consistently produced. Essential oils have, thus, emerged as candidates for this purpose; these are mixtures of organic compounds, i.e., metabolites, that are produced by plants and are capable of acting as bactericides, antioxidants, antivirals, fungicides, and insecticides. The antioxidant and antimicrobial properties of essential oils from species of the genera *Citrus* and *Xylopia* offer alternative phytogenic additives to replace growth promoters in feed while maintaining the performance and quality of the animal origin product. So, the present study aimed to determine the effects of the addition of essential oils from the fruits of *C. sinensis* and *X. aromata* on the diets of broiler chickens as phytogenic additives in their feed.

**Abstract:**

This study aimed to evaluate the effects of essential oils from the fruits of *Citrus sinensis* and *Xylopia aromatica*, included in broiler feed, on blood parameters, the biometrics of digestive organs, bone analyses, and the biochemical profiles of the viscera, as well as the histomorphometry of the small intestine. In this study, 180 one-day-old male chicks of the Cobb 500 strain were fed a corn and soybean meal over three treatments and six replications, and the experimental design was completely randomized. The data were subjected to an analysis of variance and a Tukey test at a 5% significance level. The effect of the experimental diets on performance, blood parameters, biometrics of the digestive organs, bone analysis, and biochemical profiles of the viscera, as well as the histomorphometry of the small intestine, were evaluated. The compounds identified in the essential oil of *X. aromatica* were sylvestrene, α-pinene, and β-pinene, while in *C. sinensis* they were limonene and myrcene. The essential oils of *C. sinensis* and *X. aromatica* had no significant effect on performance at 14 days. The effects of the presence of the essential oils of *C. sinensis* and *X. aromatica* on the response were beneficial: there were reductions in liver lipids, cholesterol, and triglycerides, and in the depths of the crypts in the jejunum of chickens. So, the essential oils from the fruits of *C. sinensis* and *X. aromatica* can be used in broiler chickens to improve the lipid profiles of birds without affecting their performance.

## 1. Introduction

The use of antibiotics selectively stimulates the growth of healthy intestinal microbiota, thus contributing to animals’ growth; however, the inclusion of this type of additive in the diet can cause problems related to the health of such animals due to resistance that develops over time, thus harming the immune system. However, the European Union’s 2006 ban on the use of antibiotics as growth promoters in poultry has increased the amount of research that has been conducted on the search for alternative phytogenic additives. However, animal nutrition plays an important role in the production of broiler birds through the use of high-quality ingredients with nutritional levels that fully meet this animal’s needs both for life and produce functions without the use of antibiotics [1,2,3,4].

Given the need to remove antibiotics from feed, studies in the area of nutrition have investigated possible natural substitutes. Among those identified are essential oils, which are lipophilic molecules with low molecular weights, consisting of mixtures of volatile organic compounds produced as secondary metabolites by plants, hydrocarbons (terpenes and sesquiterpenes), or oxygenated compounds (alcohols, ethers, aldehydes, ketones, lactones, and phenols) [5].

Because the diversity of metabolites is immense, with approximately 320,000 secondary metabolites existing, and a single plant has a biosynthesis capacity of up to 25,000 compounds [6,7], essential oils have been widely used as bactericides, antioxidants, antivirals, fungicides, insecticides, acaricides, in antiparasitic activities, expectorants, and for their anticancer and cytotoxic properties [5,8,9,10].

As natural products, they have also been studied as phytogenic additives, since they contribute to the better use of nutrients and, consequently, improve performance, as they stimulate the appetite and the secretion of endogenous digestive enzymes, as well as improving the integrity of the intestinal epithelium and exhibiting antimicrobial, coccidiostatic, and anthelmintic activities [11,12,13,14,15,16,17]. 

The biological activities of the species stand out in the *Citrus* (Rutaceae) genus, in which studies have highlighted the antioxidant activity with the free radical reduction 2,2-diphenyl-1-picrilhhydrazyl (DPPH) using essential oils of *Citrus sinensis* (orange) and *Citrus latifolia*. In addition to a reduction in reactive oxygen species, the antioxidant potential of the essential oils of fruit peels of *Citrus sinensis, Citrus reticulata*, and *Citrus paradisi*, as well as their antimicrobial activity against strains of *Staphylococcus aureus, Escherichia coli*, and *Salmonella typhi*, among other pathogenic microorganisms (EROS), were highlighted [18,19].

Costa et al. [20] identified the microbiological, antioxidant, and antitumor potentials of the essential oil of *Xylopia laevigata*; Konan et al. [21] highlighted the antioxidant activity of the essential oil of *Xylopia aethiopica*; Tegang et al. [22] identified excellent antifungal and antioxidant potentials of the essential oil of *X. aethiopica*, with free radical (DPH) scavenging activity of up to 72%, highlighting the potential of this species of the genus; Wouatsa et al. [23] identified a moderate biological activity for the essential oil of *X. aethiopica* on *Bacillus subtilis*, *E. coli, Micrococcus luteus, S. aureus, Pseudomonas aeruginosa, Raoultella planticola,* and *Salmonella typhimurium*. In addition to the *Citrus* genus, the *Xylopia* genus also stands out for its antioxidant and bactericidal activities.

The potential of essential oils for their antioxidant, antimicrobial, antiparasitic, and immunomodulatory activities, among others, which points to the benefits of their use as phytogenic additives in poultry feed, has been reported in the literature [24,25]. Wouatsa et al. [23] evaluated the benefits of using the essential oils of lemon, orange, and bergamot peels as phytogenic additives, obtaining promising results regarding performance, histomorphology of the jejunum, and reduction in intestinal microbiota in broilers. Krishan et al. [26] also evaluated the effect of a natural mixture of the essential oils of basil, caraway, laurel, lemon, oregano, sage, tea, and thyme as phytogenic additives in poultry feed, obtaining an improvement in weight gain and carcass quality in broilers. Erhan et al. [27] included a mixture of essential oils in chicken feed, including oleoresin from carvacrol, cinnamaldehyde, and capsicum, checking the antioxidant potential of these essential oils, improving liver function, and increasing the concentration of carotenoids and coenzyme Q10.

Antioxidant and antimicrobial are properties of essential oils of species of the genera Citrus and *Xylopia* lead to future research on the use of these natural products as alternatives as phytogenic additives to replace growth promoters while maintaining performance and quality of the animal product. 

The *C. sinensis* essential oils can be obtained through the use of waste from the juice industry, being a great alternative for reusing raw materials. Furthermore, the use of *X. aromatica* essential oils added to animal feed has not yet been reported in the literature, highlighting the innovative nature of this research.

Thus, the objective of the present study was to determine the effects of adding essential oils from the fruits of *C. sinensis* and *X. aromatica* in broiler diets on performance, biometrics, biochemistry, and duodenal histomorphology in broilers from 1 to 14 days.

## 2. Materials and Methods

### 2.1. Extraction and Chemical Analysis of Essential Oils

In the municipality of Iporá—GO, the fruits of *X. aromatica* were collected from March to May 2017 in a cerrado region while the residues of *C. sinensis* fruits were provided by the juice company Naturanja after pulp extraction.

The collected fruits were crushed in a blender and submitted to hydrodistillation process in a Clevenger-type apparatus for a period of 3 h. Then, the hydrolate was stored in a freezer (−20 °C). After freezing, the aqueous phase was separated from the organic phase to obtain the essential oils, which underwent and chemical analyses and biological tests.

The quantification of chemical constituents of the essential oil was carried out using a GCMS System (TQ8030 Shimadzu) in the Laboratory of Chemistry of Natural Products of the Federal University of São Carlos (UFSCar), in a fused-silica capillary column RTX-5MS (30 m × 0.25 mm id, 0.25 μm film thickness, Restek) using ultra-high purity helium as a carrier gas at a flow rate of 3.0 mL/min. The mass spectrometer was operated in the electron impact mode (EI) at 70 eV, scanning at a range of 43–550 m/z. The ion source temperature was set at 230 °C.

GCMS Real Time Analysis^®^ Software n.136 (P/N: S223-57119) (2017) was used for the analysis of data separation. The temperature was initially kept at 60 °C for 3 min, followed by an increase of 3 °C/min until reaching 200 °C. Next, the temperature was programmed to increase by 15 °C/min until reaching 280 °C, which was maintained for 1 min. The apparatus settings were as follows: injection temperature—230 °C; detection temperature—300 °C; injection pressure—57.4 KPa; splitless ratio—150; detection range of the mass spectrometer—43–550 m/z; start time (cut time of the solvent)—3.0 min; flow—3 mL/min.

The oil components were identified based on the Kovats retention index, calculated in relation to the retention times of a homologous series of n-alkanes (C-7 to C-40), and based on the fragmentation pattern observed in mass spectra by comparison with the literature data and the Nist spectra [28,29].

### 2.2. Biological Assay

The experiment was carried out at the Poultry Sector of Instituto Federal Goiano—Campus of Rio Verde and at the Animal Nutrition and Biochemistry and Animal Metabolism Laboratories from August to December 2018. 

Before the arrival of the batch, the usual rules for both the shed and the batteries were followed. The period of cleaning and disinfection of the facilities (screens, curtains, floor, outdoor area, and equipment) lasted seven days; two of those days were used for cleaning and five of them were used for a sanitary break (since the installation does not maintain intermittent production), with the spraying of quaternary ammonia and glutaraldehyde-based disinfectants. 

#### 2.2.1. Broilers Farming

The experimental design was completely randomized with three treatments and six replications comprising 10 birds each. The treatments were: control, with corn and soybean meal; corn, soybean meal, and 200 ppm of *C. sinensis* essential oil; corn, soybean meal, and 200 ppm of *X. aromatica* essential oil. The inclusion of 200 ppm of essential oils was defined after a bacteriological test. A total of 180 male 1-day-old Cobb 500 chicks were used, with initial weight of ±45.3 g, housed in galvanized wire cages with dimensions 0.90 m × 0.60 m × 0.45 m. 

Bird feed and water were provided ad libitum throughout the experimental period and each cage had a trough-type feeder and drinker. The light program was adopted for 24 h, considering natural and artificial light. Diets were prepared at Instituto Federal Goiano according to the following phases: pre-initial (1 to 7 days) and initial (7 to 14 days) following the nutritional recommendations of [30] (Table 1).

Average maximum and minimum temperatures and relative air humidity measured throughout the experiment are shown in Table 2.

#### 2.2.2. Performance Evaluation

Feed and poultry were weighed throughout the phases of 1–7 days, 8–14 days, and 1–14 days to assess weight gain, feed intake, feed conversion. Each repetition of each treatment was weighed so an average weight among the animals could be established. Euthanasia was performed on the bird presenting the average weight of the repetition.

#### 2.2.3. Morphometry of the Gastrointestinal Tract and Tibia

On the 7th and 14th days, one bird from each repetition was separated for fasting and around 8 h after they were euthanized by cervical dislocation. During the necropsy, the viscera that make up the gastrointestinal tract (GIT) were removed; Those were weighed and measured according to the length of the GIT, as measured by the size of the GIT from the insertion of the esophagus into the oropharynx until the communication of the large intestine with the cloaca; Then, the weight of the organs was measured separately as follows: esophagus and crop; pro-ventricle and gizzard (with remaining content); pancreas, after its separation from the duodenal loop; small intestine (SI), portion comprising the end of the muscular stomach to the beginning of the caecum; large intestine (LI), represented by the weight of caecum, colon, and rectum; liver, given by liver weight without gallbladder [31]. 

Results were converted to relative weights according to Equation (1):Relative Organ Weight (ROW) = (Organ Weight/Body Weight) × 100(1)

Tibial bones were cleaned from all adherent tissue, weighed on a precision scale, and their length and diameter were measured with a digital caliper to determine the Seedor index (bone mass index). 

Results were converted to relative weights according to Equation (2):Bone Mass Index (BMI) = Organ Weight/Organ Length(2)

#### 2.2.4. Metabolizability Apparent Analysis

Excreta were collected for five days in trays and stored inside adequately identified plastic bags in a freezer until laboratory analysis after a period of adaptation of three days. Ferric oxide at 1% was used as a fecal indicator in the rations corresponding to the first and last excreta collection to determine the beginning and end of collections. The poultry feed and excreta were thawed, weighed, homogenized, and pre-dried in a forced circulation oven at 55 °C. After reaching equilibrium with room temperature and humidity, samples were again weighed for humidity calculation. The excreta aliquots were ground in a Willey type mill using a 2 mm sieve for the analysis of dry matter (DM), crude protein (CP), and ether extract (EE), following the methodology of [32].

The metabolizability calculations for the determination of dry matter (DM), crude protein (CP), and ether extract (EE) are expressed in Equations (3)–(5):DM (%) = ((DM feed − DM excreta)/DM feed) × 100(3)
CP (%) = ((CP feed − CP excreta)/CP feed) × 100(4)
EE (%) = ((EE feed − EE excreta)/EE feed) × 100(5)

#### 2.2.5. Serum, Liver, and Pancreas Biochemical Profiles

The blood of euthanized animals was collected by cardiac puncture, centrifuged at 5000 rpm for 10 min, and the serum was frozen for further analysis using commercial DOLES^®^ kits. The biochemical evaluation of the serum was performed according to [33] with adaptations, indicating levels of calcium (Ca), phosphorus (P), total proteins (TP), alkaline phosphatase (AP), triglycerides (T), cholesterol (Col), lipases (Lip), amylase (Am), and the enzymes, glutamate-oxaloacetate transaminase (TGO), and glutamate-pyruvate transaminase (TGP).

Also, the livers and pancreases of the birds were removed and frozen for further biochemical analysis. One gram of each sample was ground and homogenized in 9 mL of distilled water. In the liver, the content of total proteins (TP), cholesterol (Col), triglycerides (T), and the enzymes glutamate-oxaloacetate transaminase (TGO) and glutamate-pyruvate transaminase (TGP) were analyzed. As for the pancreas, total protein content (TP), lipases (Lip), and amylases (Am) were evaluated [33], with commercial DOLES^®^ kits.

#### 2.2.6. Intestinal Histomorphometry

Histomorphometric analyses of the duodenal mucosa were performed using images obtained in 4 × magnifications with the aid of an optical microscope and analyzed using Image-Pro Plus^®^ F5 Software 7.0 (2004–2023). The studied variables were the height of intestinal villi, the depth of the crypts and the villus/crypt ratio. 

Villus measurements were made from the basal region, coinciding with the upper portion of the crypts up to the apex of the villi. The measurements of the depth of the crypts were taken from the basal region of the villi to its delimitation with the muscularis mucosa; the villi and crypts of the duodenum were also evaluated microscopically after slaughter using a methodology adapted from [34]. Intestinal segments of approximately 4.0 cm length were carefully collected and washed immediately in distilled water, identified, and stored in a 10% formaldehyde solution for 24 h until the slides were prepared.

For the assembly of slides, intestinal cuts were dehydrated in an increasing series of ethanol, diaphanized in xylol, and included in paraffin. Afterwards, 6.0 μm thick multi-series cuts were made, placed on glass slides, stained in hematoxylin–eosin, and covered with glass coverslips.

### 2.3. Statistical Analysis

The experiments consisted of three treatments and six replications, and the experimental design was completely randomized. Data were subjected to analysis of variance (ANOVA), and treatment means were evaluated by Tukey test at 5% significance by SISVAR^®^ statistical software (5.6), 2019 [35].

## 3. Results

### 3.1. Chemical Composition of Essential Oils

The hydrodistillation of *X. aromatica* and *C. sinensis* fruits provided translucent oils, less dense than water, with an average yield of 1.4% ± 0.01 and 0.5% ± 0.01. 

In the essential oil of *X. aromatica* fruits, the main compounds found were Sylvestrene (63.56%), α-Pinene (18.00%), β-Pinene (10.88%), Myrcene (2.73%), α-Phellandrene (2.20%), and Sabinene (2.00%), while in the essential oil of *C. sinensis* fruits, seven chemical compounds were identified, with the majority being Limonene (95.64%) and Myrcene (1.86%) (Table 3).

### 3.2. Performance Analysis

The performance results of the broilers fed in the period from 1 to 7 days and 8 to 14 days and total period of the experiment from 1 to 14 days with diets containing 200 ppm of *X. aromatica* and *C. sinensis* essential oils are shown in Table 4. The supplementation of *X. aromatica* and *C. sinensis* essential oils in the diet of the broilers did not result in a difference in weight gain, feed consumption, or feed conversion.

### 3.3. Nutrient Metabolism Analysis

The metabolism coefficients of crude protein, ether extract, and dry matter in broilers fed with diets containing 200 ppm of *C. sinensis* and *X. aromatica* essential oils showed no significant difference in the pre-initial phase. However, in the initial phase, *C. sinensis* and *X. aromatica* essential oils presented higher metabolism coefficients of dry matter, crude protein, and ether extract than control treatment (*p* < 0.05) (Table 5).

### 3.4. Morphometry of the Gastrointestinal Tract and Tibia

The results of the gastrointestinal tract morphometry are shown in Table 6. No significant difference was found between the biometrics of the organs that make up the gastrointestinal tract, except for the weight of Bursa of Fabricius (Table 6). 

In seven days, the Bursa of Fabricius of broilers fed with control treatment and *C. sinensis* essential oil were larger than the Bursa of Fabricius of broilers fed with *X. aromatica* essential oil. In 14 days, the Bursa of Fabricius of broilers fed with control treatment was larger than the Bursa of Fabricius of broilers fed with *C. sinensis* and *X. aromatica* essential oils.

In the evaluation of tibial morphometry (Table 7), no significant difference was found between treatments for weight, diameter, and bone mass index at 7 and 14 days.

### 3.5. Biochemical Profiles of Serum, Liver, and Pancreas

The results of serum biochemical profile of pre-initial and initial phases are shown in Table 8. No significant difference was found between treatments for content of phosphorus, calcium, total proteins (TP), alkaline phosphatase (AP), cholesterol, and amylase in the pre-initial phase (Table 8).

However, in this phase, a significant decrease was observed for the contents of triglycerides, glutamate-oxaloacetate transaminase (TGO), and glutamate-pyruvate transaminase (TGP) in blood serum in the treatments with *C. sinensis* and *X. aromatica* essential oils, when compared to the control treatment. Also, in the serum, the levels of lipases were lower in the treatment with *C. sinensis* than in the one with *X. aromatica* essential oil and control.

In the initial phase, no significant difference was observed for cholesterol content and for the enzymes alkaline phosphatase (AP), glutamate-oxaloacetate transaminase (TGO), and glutamate-pyruvate transaminase (TGP). However, an increase in serum phosphorus levels for birds fed with *C. sinensis* essential oil and a decrease in calcium levels in birds fed with *X. aromatica* essential oil were observed. In birds fed with *X. aromatica* essential oil, a significant increase was present in the protein content (TP) and a decrease was shown in lipase and amylase content in the blood serum. Also, birds fed with *C. sinensis* and *X. aromatica* essential oils had a significant reduction in the content of triglycerides when compared to the control treatment.

The results of the biochemical analysis of liver are shown in Table 9. 

At seven days of age, higher levels of total proteins, triglycerides, and glutamate-pyruvate transaminase enzyme were observed in the liver of broilers fed with control diet than in the liver of broilers fed with essential oils. In the pre-initial phase, the levels of cholesterol (Col) content in the liver of broilers fed with *X. aromatica* essential oil were significantly lower.

In the initial phase, birds fed with a diet containing *C. sinensis* and *X. aromatica* essential oil had lower cholesterol content than birds fed with the control diet, and birds fed with *C. sinensis* essential oil had a significant reduction in the content of triglycerides. The other analyzed parameters were not significantly influenced.

In the pancreas, the analysis of total protein and lipase content in the pre-initial and initial phases showed no significant difference between treatments (Table 10).

In the initial phase, higher levels of pancreatic amylase were observed in broilers fed with 200 ppm of *C. sinensis* and *X. aromatica* than in broilers fed with control diet.

### 3.6. Intestinal Histomorphometry

The results of the villus height, depth of the crypts, and villus/crypt ratio of the duodenum of broilers from 1 to 14 days old are shown in Table 11.

In the pre-initial phase, no significant difference was observed for villus height of the in the duodenum, but the depth of crypts was significantly greater in the control treatment than in the other treatments, which significantly reduced the villus/crypt ratio for the control diet. In the initial phase, the villus height of duodenum was significantly higher in broilers fed with *C. sinensis* and *X. aromatica* essential oils, respectively, while the depth of crypts was lower in these treatments than in broilers fed with control diet. However, no significant difference in the villus/crypt ratio between treatments at this stage.

## 4. Discussion

The *C. sinensis* essential oil has Limonene as the major compound and the essential oil of species belonging to *Xylopia* genus has the major compounds as Sylvestrene, α-Pinene and β-Pinene. These major compounds were also identified by other authors for species of *Citrus* [19] and *Xylopia* general [22,36], both from Rutaceae family. The species of this family are extensively studied, mainly the *Citrus* genus, which includes lemons, oranges, and tangerines, whose major compound is Limonene [37,38]. As for *Xylopia* genus, it is a little less studied, as it comprises less-known wild plants, and α-Pinene, β-Pinene, and Myrcene are among the major compounds of its fruits [20,39]. Despite presenting similar compounds, none of those studies mentioned the presence of Sylvestrene, the major compound identified here. Jamwa et al. [40] identified Sylvestrene among the major compounds of the essential oil of *Zanthoxylum caribaeum* (Rutaceae) leaves; the authors of [41] also identified the compound Sylvestrene among the majority of compounds in the essential oil of *Z. Bungeanum* (Rutaceae) from different regions of China.

In the present study, supplementation with *C. sinensis* and *X. aromatica* essential oils did not lead to significant results on animal performance. Similar results were reported by [42], who found there was no significant effect on the performance of broilers treated with essential oil of Mexican oregano in 7, 14, and 21 days of the experiment.

However, several authors report the positive impact of essential oils on broiler performance [43,44]. Some authors observed that supplementation with essential oils of orange, lemon, and bergamot peel significantly improved the feed conversion rate in broilers [26]. Al-Yasiry et al. [45] found better weight gain in birds fed with *Boswellia serrataresina* essential oil (Indian tree popularly known as frankincense) from 1 to 42 days. Chalghoumi et al. [46] found no significant effect in the initial phase (8–14 days) in treatments with essential oil of garlic and *cinnamon* but highlighted an improvement in performance after 35 days of experiment. Wade et al. [47] reported an improvement in the performance of broilers fed with thyme essential oil, reporting greater gain in body weight, better feed conversion, viability, and profit in the production of broilers. El-Latif et al. [48] observed improvement in the performance of broilers fed with essential oils of *Rosmarinus officinalis* (rosemary) and *Allium sativum* (garlic) for 42 days, in addition to noticing a stimulation of innate immunity by increasing the phagocytic capacity of heterophile.

Differentiation in feed consumption was expected, as essential oils stimulate appetite and release digestive enzymes, in addition to increasing nutrient absorption by increasing villi and decreasing crypts [12]. However, in the present test, there was no evidence of the effects of the essential oils of *C. sinensis* and *X. aromatica* and, consequently, of their major compounds on the performance of birds.

The metabolization of nutrients showed a significant effect at 14 days, although the feed consumption was not affected, there was greater use of nutrients. Catalan et al. [49] and Yang et al. [50] reported that essential oils improve palatability, stimulating appetite, and increasing food intake, influencing the speed of passage of food through the gastrointestinal tract, increasing the secretion of saliva, bile and mucus, and increasing enzyme activity. The *X. aromatica* essential oil has Sylvestrene, α-Pinene, and β-Pinene among its major chemical compounds, while the *C. sinensis* essential oil had an abundance of limonene, compounds that may have influenced both palatability and hydrolysis of nutrients present in the feed, which did not influence the animal performance. The metabolism coefficient reflects the digestibility of nutrients; that is, an increase in that coefficient indicates greater absorption of nutrients from a diet [51]. In this sense, it is possible to observe that a greater absorption of these nutrients was shown in diets with essential oils concerning the control group in this phase of animal development, at 14 days of age.

Amad et al. [52] observed that the metabolizability of crude protein, ether extract, calcium, and phosphorus was significantly higher in birds fed with diets containing essential oils of thyme and anise than in the control group. According to [53,54], one of the reasons why essential oils improve nutrient absorption may be the fact essential oils reduce the bacterial load by acidifying the intestinal lumen and, thereby, reduce the competition of intestinal bacteria with the host for energy supplying nutrients. For this reason, in the present work, better metabolism coefficients were observed for crude protein, dry matter, and ether extract when *C. sinensis* and *X. aromatica* essential oils were added.

The addition of *C. sinensis* and *X. aromatica* essential oils in the diet of broilers did not influence the biometrics of the gastrointestinal tract. The data in the literature on the effect of essential oils on the biometrics of the gastrointestinal tract of birds is still very controversial. Contrary to the present study, [55] found that quails that received thyme essential oil as a phytogenic additive had a larger intestine length and weight than the control treatment. Çabuk et al. [56] reported that the blend of essential oils from *Origanum* sp., *Laurus nobilis* L., *Salvia triloba* L., *Myrtus communis, Foeniculum vulgare,* and *Citrus* sp. did not result in a significant difference in organ biometrics, whereas [52] observed an increase in the relative weight of the liver in broilers from the inclusion of essential oils. The authors of [57] found a reduction in proventriculus and gizzard and an increase in the length of the duodenum in broilers fed with a diet supplemented with 3% of *Boswellia serrata*, resin, which reflected in better use of nutrients. Contradictorily, [58] found no effects between treatments with *pepper* oil on the relative weight of the organs, which is also in agreement with the results of the present study.

The size of the bursa of birds fed with *X. aromatica* essential oil was smaller than that of birds fed with the control diet at seven days of age. At 14 days, birds fed with diets containing *C. sinensis* and *X. aromatica* essential oils had bursa with small size, showing that the essential oil can have a positive effect on animal immunity, acting on the reduction in lymphocyte production. Knowledge of the typical morphology of primary and secondary lymphoid organs and tissues is an essential morphometric analytical tool for determining the intensity of the immune response in these sites [59]. The Bursa of Fabricius is a lymphoepithelial organ found only in birds. It is a rounded pouch located just above the cloaca and reaches its largest size about one or two weeks after hatching and then decreases as the bird ages; it is hardly identified in older birds [60,61,62,63]. Tizard et al. [64] state that Bursa of Fabricius is a primary lymphoid organ that functions as a place for maturation and differentiation of the cells that make up the antibody-producing system such as B and T lymphocytes, despite the involution with age.

Histomorphometric studies of Bursa of Fabricius in birds subjected to stress conditions concluded that stress affects the development of bursa with increasing size; thus, the weight of lymphoid organs such as Bursa of Fabricius, for example, reflects the body’s ability to produce lymphoid cells during the immune response [65]. 

The literature presents different data regarding the effects of essential oils on lymphoid organs. Mohammad et al. [66] evaluated the addition of *Satureja khuzistanica* essential oil in the diet of broilers and noted no significant difference in the development of lymphoid organs such as Bursa, thymus, and spleen at 21 and 42 days, in the different doses tested and in the control treatment. El-Latif et al. [48] also found no significant difference in the weight of the Bursa between control and treatments containing different doses of essential oils of *Rosmarinus officinalis* (rosemary) and *Allium sativum* (garlic). Yang et al. [50] found that the mixture of organic acids and essential oils had no significant effect on the performance and size of Bursa of the 21- and 42-day-old broilers. Contradictorily, when analyzing the immune-stimulating effect of essential oils of *peppermint* and *eucalyptus* added to water at a dose of 0.25 mL/L [67] found an increase in the weight of Bursa of chickens treated with those essential oils and concluded they are capable of implementing an immune response in broilers.

Bone mineral density can be measured by the Seedor index that represents the weight/length ratio of the organ [68]. The Seedor index of tibia and its diameter was not influenced by the diet containing *C. sinensis* and *X. aromatica* essential oils. Murakami et al. [69] studied the effect of adding flaxseed oil to the feed of broilers and observed a positive influence of these on the bone development of the animal, verifying greater mineralization of the femur and tibia bones. The addition of lipids in broiler diets is known to promote a significant reduction in the rate of bone calcification [70]. However, there are no records in the literature on the influence of essential oils and their major compounds on bone metabolism.

Calcium (Ca) and phosphorus (P) ions are essential to maintain animal homeostasis in the 2:1 ratio and are related to bone formation [71]. However, the 2:1 ratio was not observed in the present experiment, not even for the control treatment. In this experiment, at 14 days, a significant differentiation in the levels of calcium and phosphorus was shown for the essential oil of *C. sinensis*, a result that is not reflected in the differentiation of bone biometrics. Cardoso-Teixeira et al. [72] observed that monoterpenes such as limonene have vasorelaxant activity due to electromechanical coupling, which promotes the influx of Ca^2+^ ions into the intracellular medium, which may have reflected in the imbalance of these ions.

The biochemical constituents of blood, liver, and pancreas reflect the physiological responses resulting from internal (age and sex) and external (food and environment) factors, providing information on the metabolism and health of animals. Thus, the feed consumption and diet composition affect blood, enzymatic, and metabolic parameters [73]. The present study analyzed the biochemical parameters of blood, liver, and pancreas, which revealed the animals’ metabolic response to the diet supplemented with essential oils of *C. sinensis* and *X. aromatica*.

The protein concentration in birds consists of 40 to 50% of albumin, in addition to transport and coagulation proteins, enzymes and hormones produced in the liver, and immunoglobulins synthesized by B lymphocytes and plasma cells [74]. This parameter was observed in the blood of broilers fed with *X. aromatica* essential oil at 14 days of age, which had levels of total proteins higher than those of the control diet. As essential oils can have immunomodulatory effect, a stimulus to the immune system can increase the bird’s ability to synthesize antibodies, represented by immunoglobulins, which would lead to an increase in serum levels of proteins such as albumin, alpha globulins (1 and 2), beta globulins, and gamma globulins, among others [75,76]. 

The primary lipids present in the blood are triglycerides and cholesterol. Triglycerides are an essential source of energy for the body and the main lipids in adipose tissue and circulate through the blood of the intestine to adipose tissues, where they are stored to supply energy to the muscles during fasting periods [77]. 

Both cholesterol and triglycerides, as they are insoluble in the blood, need an adjuvant to be transported within the bloodstream, such as HDL (high-density lipoprotein), LDL (low-density lipoprotein), IDL (intermediate-density lipoprotein), VLDL (very-low-density lipoprotein), and chylomicrons. VLDLs are synthesized in the liver and transport triglycerides from the liver to peripheral tissues. LDL and IDL are synthesized in plasma by the action of lipoprotein lipase. As the VLDL deposit their triglycerides, there is an increase in density due to the release of triacylglycerol fatty acid. These fatty acids enter the cell where they are re-esterified for storage or are metabolized for energy. The glycerol of triglyceride returns to the liver, where it is metabolized [78,79].

Therefore, the concentration of circulating triglycerides reflects the balance between its intestinal absorption, its synthesis, and secretion in hepatocytes, and its absorption in adipose tissue, influenced by the fat content in the diet and the production of hormones [80,81]. In this work, triglycerides in blood serum were reduced in treatments with *C. sinensis* and *X. aromatica* essential oils, at 7 and 14 days, as well as in the liver at 7 days, being a good indicator of animal health. Another indication of the animal’s health was verified with the reduction in cholesterol in the liver of chickens fed with essential oils of *C. sinensis* and *X. aromatica* at 14 days.

The use of natural compounds as essential oils in the feed can lead to hypocholesterolemia, by inhibiting the regulatory enzyme of cholesterol synthesis, 3-hydroxy-3-methylglutaryl coenzyme A (HMG-CoA) reductase, produced in the liver [78,82]. The inhibition of cholesterol synthesis requires two regulators that modulate HMG-CoA reductase activity, LDL-derived cholesterol, and non-steroidal products derived from mevalonic acid [78,83]. The active principles of some essential oils like Timol, Carvacrol, and Borneol can cause hypocholesterolemia in broilers inducing non-steroidal regulatory products like mevalonate that inhibit HMG-CoA reductase [84,85]. 

The mechanism of action of limonene has been studied by [86], who report its potential to inhibit 70% of HMG-CoA reductase at a post-transcriptional level, thereby interfering with cholesterol synthesis. Limonene is metabolized in the liver and transformed into perillyl alcohol, perillyl aldehyde, and perillyl acid, all of which can prevent protein isoprenylation by interfering with the isoprenoid pathway [87]. These data justify the benefits of adding the essential oils of *C. sinensis,* rich in Limonene, and of *X. aromatica* in the feed of broilers. However, no reports were found in the literature on the mechanism of action of the major compounds of *X. aromatica* essential oil.

Literature data on the effects of natural products such as essential oils on biochemical parameters are very different. The data obtained here corroborate those found by [88], who found a reduction in cholesterol and triglycerides in quail eggs fed with 0.5% saffron (*Curcuma longa*). Gerzilov et al. [89] also indicated a significant reduction in the concentrations of triglycerides and total cholesterol in the blood of ducks fed with a mixture of herbs, a reduction in cholesterol and triglyceride levels with the addition of turmeric in quail diets at doses starting from 0.05 g/100 g of feed, highlighting the ability to inhibit the synthesis of triglycerides in liver cells. According to the authors, the expression of genes involved in energy metabolism may have been influenced by turmeric, resulting in reduced expression, decreasing the accumulation of fats in the blood and tissues, with decreased intracellular levels of lipids [90,91]. Domingues et al. [58] also found a reduction in triglyceride levels in broilers fed with Piper cubeba. Manan et al. [92] observed a decrease in triglyceride levels when supplementing the diet of broilers with medicinal plants. Contradictorily, some authors have not verified the effect of rosemary and garlic essential oils on broiler performance, besides causing an increase in serum triglyceride levels, total cholesterol, low-density lipoprotein (LDL), and high-density lipoprotein (HDL) [48].

In the present experiment, blood lipase levels were lower at seven days in broilers fed with essential oil of *C. sinensis*, which was not observed at 14 days. In this initial phase, the levels of lipase and amylase were slightly lower in broilers fed with essential oil of *X. aromatica* than broilers fed with other treatments. However, such variations were not observed in the biochemical analysis of the pancreas. In broilers, essential oils can also stimulate the activity of certain digestive and pancreatic enzymes [93]. Another explanation for the increase in enzyme activity and pancreatic secretion is the increase in the relative weight of pancreas that those oils could induce [93,94]. However, such a change was not observed in the present study.

The levels of glutamate-oxaloacetate transaminase (TGO) and glutamate-pyruvate transaminase (TGP) enzymes in the blood were lower in broilers fed with essential oils of *C. sinensis* and *X. aromatica* than broilers fed with control diet at 7 days, a result that was not observed at 14 days. In the liver, only the levels of glutamate-pyruvate transaminase (TGP) were lower in broilers fed with essential oils of *C. sinensis* and *X. aromatica* than broilers fed with control diet at seven days. The glutamate-oxaloacetate transaminase (TGO) and glutamate-pyruvate transaminase (TGP) enzymes are essential in the diagnosis of liver damage caused by drugs or infections, since after these events several enzymes, including aminotransferases, leak from the injured cells and pass into the blood [95]. However, such lesions were not observed in the present study. 

Borsa et al. [96] defined the following values for glutamate-pyruvate transaminase (TGP) in serum: 2–13 and 9–22 (U/dL) for birds at 7 and 14 days, respectively. Minafra et al. [97] found values of 203–267 (U/dL) at 7 days, with values of 251–375 (U/dL) at 14 days for glutamate-oxaloacetate transaminase (TGO) levels; they also found values around 73.3–79.5 (U/dL) at 7 days, with 72.2–75.6 (U/dL) at 14 days for glutamate-pyruvate transaminase levels in the liver tissue of chickens fed with diets supplemented with glutamic acid and vitamin K. These results are divergent in relation to the biochemical parameters of normality for this enzyme in liver tissue.

In another study, the addition of a mixture of essential oils containing Timol and Carvacrol in the diet of ducks also did not change the biochemical parameters regarding the levels of total proteins, glycemia, calcium, triglycerides, cholesterol, and the enzymes glutamate-oxaloacetate transaminase and glutamate-pyruvate transaminase, but they significantly improved phagocytic activity when compared to the control [97].

In the present study, the addition of *C. sinensis* and *X. aromatica* essential oils to the broiler feed positively influenced the intestinal health of animals, promoting an increase in the villus height and depth of crypts, mainly in the initial phase of animal development, which evidences the maintenance of the intestinal integrity of the animal. According to El-Katcha et al. [98], the villus height and the depth of crypts are considered indicators of proper intestinal development.

The results found in the present study corroborate the literature data regarding the effect of essential oils on the intestine histomorphometry of broilers. Kuzmuk et al. [99] also found greater villus height and less depth in the ileum crypts of 21- and 42-day-old broilers fed with essential oils of oregano (*Origanum onites*), cloves (*Syzygium aromaticum*), and cumin (*Cuminum cyminum*). Basmacioğlu-Malayoğlu et al. [100,101], found greater villus height and depth of crypts in the jejunum of chickens fed with a mixture of organic acids and essential oil at 42 days of age. Yang et al. [50] also found greater villus height and depth of duodenal crypts of birds fed with a mixture of sorbic acid, fumaric acid, and thymol at 42 days of age, highlighting the potential of using essential oils and their active principles in improving the health of birds.

## 5. Conclusions

The results suggest that the essential oils of *C. sinensis* and *X. aromatica* in the tested dose (200 ppm/kg of feed) can be used in diets for broilers up to 14 days of age to reduce cholesterol and triglyceride levels.

## Figures and Tables

**Table 1 animals-13-03326-t001:** Centesimal composition and calculated nutritional levels of the diets: control, and with the addition of 200 ppm of essential oils of *C. sinensis* or *X. aromatica*.

Ingredients (g/kg)	Pre-Initial	Initial
Yellow Corn 8.58%	55.3000	56.0200
Soybean meal 46%	39.3706	37.9300
Soybean oil	0.8000	1.7000
Limestone	2.2000	1.1200
Dicalcium phosphate	0.0648	1.2500
Premix Vit/Min	1.0000 *	1.0000 **
DL- Methionine	0.2645	0.2200
L- Lysine	0.3000	0.2000
L- Threonine	0.1944	0.0700
Common salt	0.5058	0.4900
Total	100.0000	100.0000
Calculated levels		
Met. energy (Kcal/kg)	3000.00	3100.00
Crude protein (%)	25.31	24.30
Digestible lysine (%)	1.36	1.31
Digestible methionine (%)	0.55	0.53
Calcium (%)	1.01	0.91
Available phosphorus (%)	0.48	0.43
Sodium (%)	0.23	0.22
Analyzed composition		
Dry Matter (%)	92.44	91.88
Crude protein (%)	26.42	26.42
Crude fiber (%)	21.86	21.03
Ether extract (%)	6.66	7.01

* Premix 1—Vitaminic Mineral (nutritional levels per kilogram of product), Methionine (Min): 290 g/kg; Iron (Min): 5000 mg/kg; Copper (Min): 1500 mg/kg; Manganese (Min): 14 g/kg; Zinc (Min): 12 g/kg; Iodine (Min): 28 mg/kg; Selenium (Min) 70 mg/kg; Vitamin A (Min): 1,500,000 IU/kg; Vitamin D3 (Min): 500,000 IU/kg; Vitamin E (Min): 3333 IU/kg; Vitamin K3 (Min): 250 mg/kg; Vitamin B1 (Min): 300 mg/kg; Vitamin B2 (Min): 1,000 mg/kg; Vitamin B6 (Min): 500 mg/kg; Vitamin B12 (Min) 3.333 mcg/kg; Niacin (Min): 6.667 mg/kg; Calcium Pantothenate (Min): 2000 mg/kg; Folic Acid (Min): 280 mg/kg Biotin (Min): 8.3 mg/kg; Choline Chloride (Min): 70 mg/kg. ** Premix 2—Vitaminic Mineral (nutritional levels per kilogram of product), Methionine (Min): 300 g/kg; Iron (Min): 6000 mg/kg; Copper (Min): 1850 mg/kg; Manganese (Min): 16.8 g/kg; Zinc (Min): 14.5 g/kg; Iodine (Min): 330 mg/kg; Selenium (Min) 84 mg/kg; Vitamin A (Min): 1,500,000 IU/kg; Vitamin D3 (Min): 500,000 IU/kg; Vitamin E (Min): 3600 IU/kg; Vitamin K3 (Min): 240 mg/kg; Vitamin B1 (Min): 300 mg/kg; Vitamin B2 (Min): 1100 mg/kg; Vitamin B6 (Min): 500 mg/kg; Vitamin B12 (Min) 3600 mcg/kg; Niacin (Min): 7000 mg/kg; Calcium Pantothenate (Min): 2000 mg/kg; Folic Acid (Min): 320 mg/kg Biotin (Min): 6 mg/kg; Choline Chloride (Min): 65 mg/kg. Inclusion of 200 ppm of *X. aromatica* and *C. sinensis* essential oils.

**Table 2 animals-13-03326-t002:** Average temperature and relative air humidity inside the shed in each production phase.

Phase	Temperature (°C)		Relative Air Humidity (%)	
	Maximum	Minimum	Maximum	Minimum
Pre-initial	29.77	18.22	76.00	58.37
Initial	27.83	17.21	67.50	58.87

**Table 3 animals-13-03326-t003:** Chemical constituents of the essential oil of *X. aromatica* (OEX) and *C. sinensis* (OEC) fruits.

Peak	R_t_ (min)	Compound	OEX ^1^ (%)	OEC ^2^ (%)	RI exp. *	RI lit. **
1	5.516	1R-(+)-α-Pinene	0.17	-	927	-
2	5.716	α-Pinene	18.00	0.53	934	932
3	6.781	Sabinene	2.00	0.32	974	969
4	6.901	β-Pinene	10.88	-	979	974
5	7.239	Myrcene	2.73	1.86	992	988
6	7.701	α-Phellandrene	2.20	0.24	1007	1002
7	8.350	ο-Cymene	0.46	-	1025	1022
8	8.505	Sylvestrene	63.56	-	1030	1030
9	13.471	Limonene	-	95.64	1036	1034
10	16.202	Linalol	-	0.55	1096	1095
11	20.608	α-Terpineol	-	0.87	1189	1186

OEX ^1^= essential oil of *Xylopia aromatica*. OEC ^2^ = essential oil of Citrus sinensis. * = retention index calculated using the Van den Dool and Kratz equation. ** = Literature retention index (ADAMS, 2007) based on the Van den Dool and Kratz equation.

**Table 4 animals-13-03326-t004:** Performance of broilers fed in the period from 1 to 7 days, 8 to 14 days, and 1 to 14 days with diets containing 200 ppm of *C. sinensis* and *X. aromatica* essential oils.

**Pre-Initial Phase (1–7 days)**					
**Variables**	**Treatments**			**Probabilities**	
	**Control**	** *C. sinensis* **	** *X. aromatica* **	***p*-Value**	**SEM**
**WG (g)**	0.105	0.107	0.100	0.078	0.0020
**FCP (g)**	0.124	0.123	0.120	0.762	0.0035
**FCV**	1.176	1.153	1.200	0.715	0.0399
**Initial Phase (8–14 days)**					
**Variables**	**Treatments**			**Probabilities**	
	**Control**	** *C. sinensis* **	** *X. aromatica* **	***p*-Value**	**SEM**
**WG (g)**	0.278	0.265	0.270	0.433	0.0065
**FCP (g)**	0.347	0.341	0.344	0.762	0.0054
**FCV**	1.251	1.289	1.275	0.462	0.0213
**Overall (1–14 days)**					
**Variables**	**Treatments**			**Probabilities**	
	**Control**	** *C. sinensis* **	** *X. aromatica* **	***p*-value**	**SEM**
**WG (g)**	0.406	0.407	0.411	0.195	0.0021
**FCP (g)**	0.506	0.520	0.501	0.502	0.0118
**FCV**	1.246	1.281	1.220	0.388	0.0304

SEM: standard error of means; WG: weight gain; FCP: feed consumption; FCV: feed conversion.

**Table 5 animals-13-03326-t005:** Metabolism coefficients of dry matter, crude protein, and ether extract in broilers fed with diets containing 200 ppm of *C. sinensis* and *X. aromatica* essential oils.

**Pre-Initial Phase (1–7 days)**					
**Variables**	**Treatments**			**Probabilities**	
	**Control**	** *C. sinensis* **	** *X. aromatica* **	***p*-Value**	**SEM**
**MCCP**	69.41	75.10	74.43	0.359	2.912
**MCDM**	65.19	66.04	68.30	0.904	5.058
**MCEE**	70.95	78.31	78.86	0.1094	2.646
**Initial Phase (8–14 days)**					
**Variables**	**Treatments**			**Probabilities**	
	**Control**	** *C. sinensis* **	** *X. aromatica* **	***p*-Value**	**SEM**
**MCCP**	81.11 b	90.87 a	91.46 a	0.0023	0.865
**MCDM**	79.57 b	88.28 a	88.63 a	0.0001	0.095
**MCEE**	84.10 b	88.45 a	87.82 a	0.0050	0.765

Means on the same row with different letters (a, b) are significantly (*p* < 0.05) different. SEM: standard error of means; MCCP: metabolism coefficient of crude protein; MCDM: metabolism coefficient of dry matter; MCEE: metabolism coefficient of ether extract.

**Table 6 animals-13-03326-t006:** Biometric evaluation of the organs of broilers in the period from 1 to 7 days and 8 to 14 days with diets containing 200 ppm of *C. sinensis* and *X. aromatica* essential oils.

**Pre-Initial Phase (1–7 days)**					
**Variables**	**Treatments**			**Probabilities**	
	**Control**	** *C. sinensis* **	** *X. aromatica* **	***p*-Value**	**SEM**
**Liver**	3.985	3.888	3.932	0.795	0.100
**Gizzard**	7.464	7.766	7.491	0.763	0.318
**Pancreas**	0.692	0.673	0.648	0.585	0.029
**LI**	1.365	1.273	1.194	0.062	0.046
**SI**	8.371	8.010	7.970	0.580	0.294
**GITW**	22.750	23.114	23.056	0.921	0.680
**EPCP**	1.329	1.283	1.269	0.586	0.042
**Bursa**	0.264 a	0.244 a	0.177 b	0.050	0.007
**Initial Phase (1–14 days)**					
**Variables**	**Treatments**			**Probabilities**	
	**Control**	** *C. sinensis* **	** *X. aromatica* **	***p*-Value**	**SEM**
**Liver**	3.405	3.435	3.416	0.986	0.132
**Gizzard**	5.648	5.071	5.109	0.103	0.197
**Pancreas**	0.549	0.532	0.528	0.660	0.016
**LI**	1.021	0.978	0.942	0.590	0.053
**SI**	5.434	5.512	5.497	0.962	0.211
**GITW**	16.979	16.463	16.390	0.506	0.380
**EPCP**	0.830	0.833	0.848	0.926	0.034
**Bursa**	0.648 a	0.485 b	0.481 b	0.050	0.025

Means on the same row with different letters (a, b) are significantly (*p* < 0.05) different. SEM: standard error of means; LI: large intestine; SI: small intestine; GITW: gastrointestinal tract weight; EPCP: esophagus and crop.

**Table 7 animals-13-03326-t007:** Bone biometric evaluation of broilers in the period from 1 to 7 days and 8 to 14 days with diets containing 200 ppm of *C. sinensis* and *X. aromatica* essential oils.

**Pre-Initial Phase (1–7 days)**					
**Variables**	**Treatments**			**Probabilities**	
	**Control**	** *C. sinensis* **	** *X. aromatica* **	***p*-Value**	**SEM**
**BMIT (g/mm)**	72.46	70.90	70.74	0.757	0.0017
**DT (mm)**	3.61	3.34	3.68	0.355	0.1634
**Initial Phase (1–14 days)**					
**Variables**	**Treatments**			**Probabilities**	
	**Control**	** *C. sinensis* **	** *X. aromatica* **	***p*-Value**	**SEM**
**BMIT (g/mm)**	152.01	141.85	148.14	0.568	0.0006
**DT (mm)**	4.74	4.99	4.99	0.648	0.2113

SEM: standard error of means; BMIT: bone mass index of the tibia; DT: diameter of the tibia.

**Table 8 animals-13-03326-t008:** Serum biochemical profile of broilers in the period from 1 to 7 days and 8 to 14 days with diets containing 200 ppm of *C. sinensis* and *X. aromatica* essential oils.

**Pre-Initial Phase (1–7 days)**					
**Variables**	**Treatments**			**Probabilities**	
	**Control**	** *C. sinensis* **	** *X. aromatica* **	***p*-Value**	**SEM**
**Phosphorus**	6.95	6.97	6.52	0.141	0.162
**Calcium**	6.38	6.74	6.39	9.030	0.240
**TP**	2.62	2.73	2.80	0.265	0.290
**Cholesterol**	125.01	123.30	118.64	0.580	4.350
**Triglycerides**	180.42 a	110.89 b	102.51 b	0.000	2.718
**Lipase**	171.17 a	89.68 b	176.22 a	0.000	3.49
**Amylase**	642.20	634.96	616.50	0.094	7.639
**AP**	68.18	65.54	67.40	0.338	1.229
**TGP**	34.93 a	16.65 b	17.08 b	0.000	0.701
**TGO**	141.05 a	106.72 b	101.14 b	0.002	6.026
**Initial Phase (1–14 days)**					
**Variables**	**Treatments**			**Probabilities**	
	**Control**	** *C. sinensis* **	** *X. aromatica* **	***p*-Value**	**SEM**
**Phosphorus**	6.12 b	6.97 a	6.12 b	0.008	0.174
**Calcium**	6.93 a	6.99 a	6.20 b	0.020	0.180
**TP**	2.66 b	2.79 b	3.25 a	0.002	0.091
**Cholesterol**	197.33	173.80	171.44	0.140	9.223
**Triglycerides**	179.75 a	110.36 b	116.49 b	0.000	2.370
**Lipase**	263.62 a	276.87 a	234.33 b	0.000	3.873
**Amylase**	624.82 a	585.74 a	427.57 b	0.000	12.14
**AP**	66.54	66.42	66.44	0.209	0.049
**TGP**	18.10	16.06	16.76	0.081	0.574
**TGO**	109.23	107.66	103.68	0.155	1.906

Means on the same row with different letters (a, b) are significantly (*p* < 0.05) different. SEM: standard error of means; total proteins (TP); alkaline phosphatase (AP); triglycerides (T); enzymes glutamate-oxaloacetate transaminase (TGO); glutamate-pyruvate transaminase (TGP).

**Table 9 animals-13-03326-t009:** Biochemical analysis in the liver of broilers in the period from 1 to 7 days and 8 to 14 days with diets containing 200 ppm of *C. sinensis* and *X. aromatica* essential oils.

**Pre-Initial Phase (1–7 days)**					
**Variables**	**Treatments**			**Probabilities**	
	**Control**	** *C. sinensis* **	** *X. aromatica* **	***p*-Value**	**SEM**
**TP**	3.33 a	2.63 b	2.89 b	0.000	0.077
**Cholesterol**	392.06 a	387.86 a	366.18 b	0.000	2.087
**Triglycerides**	298.18 a	240.51 b	232.84 b	0.000	2.508
**TGP**	63.10 a	41.03 b	43.21 b	0.000	1.329
**TGO**	209.01	198.52	210.82	0.150	16.960
**Initial Phase (1–14 days)**					
**Variables**	**Treatments**			**Probabilities**	
	**Control**	** *C. sinensis* **	** *X. aromatica* **	***p*-Value**	**SEM**
**TP**	2.80	2.74	3.01	0.053	0.012
**Cholesterol**	376.1 a	364.60 b	364.99 b	0.000	1.538
**Triglycerides**	215.46 a	198.80 b	213.72 a	0.000	1.971
**TGP**	41.84	42.12	43.04	0.721	1.084
**TGO**	182.55	197.90	192.76	0.171	5.374

Means on the same row with different letters (a, b) are significantly (*p* < 0.05) different. SEM: standard error of means; total proteins (TP); cholesterol, triglycerides, glutamate-oxaloacetate transaminase (TGO); glutamate-pyruvate transaminase (TGP).

**Table 10 animals-13-03326-t010:** Biochemical analysis of the pancreas of broilers in the period from 1 to 7 days and 8 to 14 days with diets containing 200 ppm of *C. sinensis* and *X. aromatica* essential oils.

**Pre-Initial Phase (1–7 days)**					
**Variables**	**Treatments**			**Probabilities**	
	**Control**	** *C. sinensis* **	** *X. aromatica* **	***p*-Value**	**SEM**
**Total proteins**	3.22	3.09	3.16	0.099	0.038
**Lipase**	163.09	152.90	158.76	0.797	10.626
**Amylase**	319.82	305.28	312.72	0.519	8.692
**Initial Phase (1–14 days)**					
**Variables**	**Treatments**			**Probabilities**	
	**Control**	** *C. sinensis* **	** *X. aromatica* **	***p*-Value**	**SEM**
**Total proteins**	3.24	3.29	3.21	0.621	0.054
**Lipase**	148.07	137.02	132.88	0.100	4.597
**Amylase**	308.57 b	332.64 a	331.08 a	0.047	6.559

Means on the same row with different letters (a, b) are significantly (*p* < 0.05) different. SEM: standard error of means.

**Table 11 animals-13-03326-t011:** Villus height, crypt depth, and villus/crypt ratio of the duodenum of broilers in the period from 1 to 7 days and 8 to 14 days with diets containing 200 ppm of *C. sinensis* and *X. aromatica* essential oils.

**Pre-Initial Phase (1–7 days)**					
**Variables**	**Treatments**			**Probabilities**	
	**Control**	** *C. sinensis* **	** *X. aromatica* **	***p*-Value**	**SEM**
**Villus (µm)**	529.82	513.36	511.04	0.485	8.745
**Crypt (µm)**	132.18 a	103.44 b	99.69 b	0.000	2.783
**Villus/crypt (µm)**	4.02 b	4.98 a	5.13 a	0.000	0.122
**Initial Phase (1–14 days)**					
**Variables**	**Treatments**			**Probabilities**	
	**Control**	** *C. sinensis* **	** *X. aromatica* **	***p*-Value**	**SEM**
**Villus (µm)**	501.99 b	557.72 a	536.72 a	0.017	7.89
**Crypt (µm)**	76.98 b	88.09 a	85.88 a	0.005	1.966
**Villus/crypt (µm)**	6.55	6.51	6.79	0.733	0.261

Means on the same row with different letters (a, b) are significantly (*p* < 0.05) different. SEM: standard error of means

## Data Availability

The data presented in this study are openly available at CHRISTOFOLI, M. Óleos essenciais de Citrus sinensis e Xylopia aromatica e sua adição em dietas de frangos de corte. 2020. 150 f. Tese (Doutorado)—Curso de Biotecnologia e Biodiversidade, Universidade Federal de Goiás, Rio Verde, 2020. Disponível em: https://files.cercomp.ufg.br/weby/up/493/o/TESE_VERS%C3%83O_FINAL_-_Marcela.pdf. Acesso em: 26 January 2022.

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
