# Peer review of "Diet of Broilers with Essential Oil from *Citrus sinensis* and *Xylopia aromatica* Fruits"

_animals, 2023, doi:10.3390/ani13213326_

Round 1

Reviewer 1 Report

Review of the manuscript animals-2614936

entitled ‘Diet of broilers witch essential oil from Citrus sinensis and Xylopia aromatica fruits’

for Animals

General comment

This manuscript is an attempt to determine the effects of adding essential oils from Citrus sinensis and Xylopia aromaticafruits in broiler diet on performance and selected physiological and anatomical parameters. Nowadays, essential oils are known to have some beneficial properties, including antimicrobial and immunomodulating, and they can be an alternative to antibiotics in broiler production, so in my opinion there is value in this idea of study.  Nevertheless, there are some important issues and limitations that prevents me to support publication of the current manuscript. The principal reason is that flaws detected in the methodology of the experiment casting doubts on the reliability of the results and hence the derived conclusions. I am afraid that it is not possible to draw conclusions, especially regarding production results, based on 2 weeks period of broilers rearing. The period of experiment was too short to determine performance results considering that broiler chickens are usually reared until 35th or even 42nd day. Moreover, the authors selected only 1 chick per group to assess morphometry of the gastrointestinal tract and tibia (by the way it is not clearly explain in methodology what was the number of samples for another parameters determination). Even if 6 replications were conducted it is not conclusive to demonstrate the effect of experimental factor on organism response after 2 measurement series, especially that in discussion the authors make very far-reaching statements and draw too certain conclusions. 

I have doubts whether the research is innovative enough to publish a work in such a reputable and internationally recognized journal as Animals. The authors themselves refer to research (lines 66-92) that have already been conducted. It should be clearly underline what innovative element this study has?

For all these reasons I do not think the paper is suitable for publication in the “Animals” as original article and I encourage the authors to treating these studies as contribution to basic research on bird physiology. To improve the manuscript the methodology should be expand with a detailed description of the number of samples, statistical analyzes (what hypotheses were verified, whether the analysis of variance was single- or multi-factorial?). Laboratory research methods should be described, the information that the DOLES kit was used is too laconic. Please, improve the English language, use formal phrases. Remember about units in the Tables (e.g., 8-10).

Detailed comments are included in the provided manuscript.

-

Author Response

I would like to thank the reviewer for his care and attention in evaluating the manuscript and for dedicating his time. I believe that all criticism is constructive to make the article more valued and meet the demands of the renowned magazine. I have tried to address all concerns appropriately and I believe the article has improved considerably.

Reviewer 2 Report

Dear Authors,

I received a paper entitled “Diet of broilers witch essential oil from Citrus sinensis and Xylopia aromatica fruits” for review. The aim of the study is to evaluate the effect of including essential oil from the fruits of Citrus sinensis and Xylopia aromatica in the broiler feed.

The topic of using phytobiotics and essential oils in poultry nutrition has been analyzed around the world for years. Phytobiotic additives themselves are of interest not only to scientists, but also to poultry producers, due to their numerous properties, which can significantly improve not only the health of broiler chicken flocks, but also reduce the use of antibiotics, improve selected production parameters and meat quality. The topic of the proposed work is interesting, the work itself is a collection of basic observations on the use of essential oils from Citrus sinensis and Xylopia aromatica fruits. The methodology used in the study is classic for this type of work, the statistical analysis is performed correctly. The work provides a lot of information, but requires thorough editing and checking in linguistic, lexical and grammatical terms. Long, complex sentences and stylistic errors make reading and understanding the work very difficult. The study itself was written in a rather chaotic manner, not very carefully, and contains a lot of punctuation errors. There are also some topics that need to be corrected or described in more details. After being edited and thoroughly checked by a native speaker, the study may meet the conditions for publication in a scientific journal such as Animals. Below are some of the detailed issues that should be corrected or supplemented:

L25 The aim of the study...to evaluate effect - what effect, effect on what? please, give the details- the aim is too general and do not give the reader the information what can exactly be found in this paper. Please, put the complete info from line 29.

L12- Capital letter-The prohibition. 12-15 please cut this one sentence into several smaller- the style and the length of this sentence make it difficult to read and understand.

L18-Capital letter- So,

L20- among others change for etc.

L20-21-please, correct the grammar and the sense of the sentence- it is difficult to understand what is the key of it.

L29-30 Please, complete the sentence- The effect on (…) was analysed? Was checked?

L33- The presence (…) on the immune – please complete the sentence – had effect, affected? Something is missed.

.The presence  of the essential oils of C. sinensis and X. aromatica on the immune response, through differentiation  of the weight of the Bursa and improvements in blood and liver lipids reduced triglycerides and  depth of the crypts, so the essential oils from the fruits of C. sinensis and X. aromatica can be used in  broiler chicken feed as phytogenic additives that promote growth.

This conclusion is false - the improvement in blood parameters, the size of the bursa and the impact on immunity cannot be clearly associated with the effect as a growth stimulator. The size of the burrow is in no way a parameter that determines growth stimulation in a broiler chicken. In general,  growth promoters are added with broiler feed to boost the overall feed efficiency and growth rate. The mere demonstration of the effect on crypts, system. immunological and selected biochemical parameters, without assessing the impact on FCR, BW, etc. cannot be a clear premise for concluding that something may or may not be a growth stimulator. The authors indicate that there was no effect on performance over a period of 1-14 days. Therefore, no effect on the basic criteria set for growth stimulants has been demonstrated.

L73- [21,22,23] etc.,who identified? Please, use the correct way of citation like Xyz et al. identified, other author [22] presented, etc. This method of notation is present throughout the entire study. Please revise it according to the animals` guidelines.

L-93-are the properties (properties- plural)

Why the experiment was designed only for day 1-14? You used Cobb500, fast growing broiler chicken which present the full performance during 42 days cycle (based on Manual for Cobb, Germany).  First two weeks in broilers is generally difficult to observe changes related to FCR, BW, FI and other stabile changes in organs weight, blood parameters etc. This period time is rather safe and birds in a farm condition do not meet any extra challenges (first week is critical when we analised the mortality related with the umblical infections or these associated with parents` flock and the hatchery quality), the real effect of feed additives as a potential growth promoter can be truly analysed for longer time period- the full cycle (35 days if shorter, 42 days with thin or without- it is related to the cobb genetic). In a conclusion section, in the intestinal descriptions the authors refer to the consistency with the results for chickens aged 21 days and/or 42 days. According to the manual, 42 days is the optimal day for ending rearing for Cobb500. This is why there are many most reliable analyzes of production parameters, but also blood tests, organ size, interleukin levels, etc. It is difficult to find a connection between the analysis of parameters on day 7/14 and those observed on day 21 or 42. The abstract concludes that the additive after 14 days of use can be administered as a growth promoter, while the final conclusions clearly emphasize the lack of impact of this preparation on production parameters. This is quite a discrepancy.

I hope these suggestions will be helpful to improve your manuscript.

Some single samples:

L12- Capital letter-The prohibition. 12-15 please cut this one sentence into several smaller- the style and the length of this sentence make it difficult to read and understand.

L18-Capital letter- So,

L20- among others change for etc.

L20-21-please, correct the grammar and the sense of the sentence- it is difficult to understand what is the key of it.

L29-30 Please, complete the sentence- The effect on (…) was analysed? Was checked?

L33- The presence (…) on the immune – please complete the sentence – had effect, affected? Something is missed.

Author Response

(The authors gave the same response as above.)

Round 2

Reviewer 1 Report

The manuscript has been fully revised point-by-point according my critical comments. The information added to the text (lines 100-103) significantly emphasizes the importance of conducted research. I believe that the paper has been sufficiently improved to warrant publication in Animals. 

I also suggest to add the conclusion that the promising results of this study should be continued in terms of the possibility of improving the performance of broiler chickens reared in the full production cycle.